# Simulation Test of The Aerodynamic Environment of A Missile-Borne Pulsed Laser Forward Detection System at High Flight Speed

**Peng Liu [1], Jian Li [1], Tuan Hua [2] and He Zhang [1,\***

[1] School of Mechanical Engineering, Nanjing University of Science and Technology, Nanjing 210094, China; njustca@njust.edu.cn (P.L.); ljian1126@njust.edu.cn (J.L.)
[2] Aviation Industry Corporation of China Jincheng Nanjing Electromechanical Hydraulic Engineering Research Center, Nanjing 211100, China; neias@neias.cn
[\*] Correspondence: hezhangz@njust.edu.cn

**Abstract:** When a missile-borne pulsed laser forward detection system flies at supersonic speed, the laser beam will be distorted by the uneven outflow field, resulting in a significant reduction in ranging accuracy. In this paper, the impact of high flight speed on a pulsed laser detection system is studied. First, a new ray tracing method with adaptive step size adjustment is proposed, which greatly improves the computational efficiency. Second, the aerodynamic environment of a munition flying at high speed is simulated by an intermittent transonic and supersonic wind tunnel to obtain the schlieren data of the flow field at various Mach numbers. The schlieren data present a shock wave structure similar to that of the simulation. In addition, the variation patterns of the pulsed laser echo waveform of the model under different aerodynamic conditions are studied to evaluate the detectability and operational stability of the laser detection system under static conditions. The test results match the simulation results well, and the two offer relatively consistent shock wave structures, which verifies the correctness and effectiveness of the flow field simulation model. The test echo waveforms are in good agreement with the simulated echo waveforms; the relative errors between the peak values of test and simulated echo waveforms at various Mach numbers do not exceed 20%, and the correlation coefficients between the test and simulated echo waveforms all exceed 0.7, indicating high correlations between the two.

**Keywords:** pulse laser; ray tracing; wind tunnel simulation; shock wave structure; correlation





## 1. Introduction

A missile-borne pulsed laser forward detection system is typically installed on the head of the munition and experiences the outer ballistic flight with the munition. The uneven flow field resulting from the extremely high flight speed disturbs the propagation of transmitting and receiving beams, thus inevitably affecting the performance and accuracy of the missile-borne pulsed laser forward detection system. When the laser beam of a pulsed lidar system passes through this flow field, optical distortions, such as offset, intensity loss, blurring, etc., will inevitably occur, which are called aero-optical effects [1–3]. When a precision-guided munition (PGM) based on optical guidance flies at high speed in the atmosphere, the optical dome interacts violently with the incoming flow, and the resulting aero-optical effect will have a great impact on optical guidance precision, which restricts the application of optical precision guidance technology in hypersonic weapons to a certain extent. Aero-optical distortion is a key problem in the development of optical guidance-based PGMs.

In aero-optics research, many breakthrough achievements often involve the internal parameters of the flow field measured through experimental methods. Among them, the Shack–Hartmann wavefront sensor (SHWS) is the most widely used wavefront sensor. The

SHWS is independent of light wavelength and is insensitive to equipment vibrations. Its dynamic measurement range is more than a hundred or thousand times that of traditional interference methods [4,5]. The Malley probe is a single-point optical phase change test tool that can measure optical phase distortion in near real-time, and the probe can also measure beam deflections at a very high frequency through a position sensor [6,7]. Compared with the SHWS, the Malley probe can directly measure optical phase changes with extremely high temporal resolution. The number of beams used and their spacing determine the spatial resolution of the Malley probe [8]. A background-oriented schlieren (BOS) achieves high spatiotemporal precision measurements by determining the flow field structure using the relationship between the density gradient field and the refractive index field and by obtaining the displacement of the light in two directions after passing through the flow field using a cross-correlation operation on the background lattice with or without flow field interference [9]. In addition to the abovementioned experimental methods, the nano-tracer planar laser scattering (NPLS) technology developed by Professor Yi can realize the high-spatiotemporal-resolution visualization of time-dependent transient flow in a section of a three-dimensional complex flow field [10]. Lee [11] investigated the aero-optical effects induced by the supersonic flow on a 2D compression ramp through wind tunnel experiments and ray tracing and determined the separate contributions of shock waves and boundary waves to the optical wavefront distortion. Hui et al. [12] proposed a ray tracing method suitable for an infrared optical dome of high-speed missiles and proposed three evaluation parameters to evaluate the changes in temperature, deformation, and refractive index of the optical dome. The Lincoln Lab [13] first defined the range-resolved lidar cross section (RRLRCS), deduced the radar cross section of a Lambertian target in a completely incoherent condition, and analyzed the effect of the incident angle and target surface size on the radar cross section. Der et al. [14] derived pulsed lidar echo signal equations under different target geometries. On this basis, Steinvall et al. [15,16] innovatively regarded the pulsed lidar as a linear system, believed that the reflection process of the laser beam on the target surface was the shock response of the linear system, and deduced the echo equations of the target with various geometric shapes. Blanquer et al. [17] performed a detailed study on the echo equation of airborne pulsed lidar and analyzed the influence of different factors on the ranging accuracy of LiDAR. Hao et al. [18] studied a comprehensive model of the echo laser pulse profile (ELPP) of an arbitrarily shaped target under the simultaneous actions of the target and atmospheric turbulence.

With the continuous development of laser detection technology, lidar and pulsed laser detection systems are increasingly used in civil and military fields, and the factors affecting the precision of pulsed laser ranging are being investigated increasingly more in depth. Amann [19] proposed that the main sources of laser ranging errors are time jitter caused by noise and time walk caused by signal amplitude. Steinvall et al. [20] studied the echo signals of lidars irradiating a cylinder on a flat plate at different incident angles and simulated and analyzed the impact of the signal-to-noise ratio of the echo signals, target geometry, reflectivity, spatial sampling, and waveform processing methods on range accuracy and lidar resolution. Oh [21] proposed a new technique to improve the accuracy of direct detection time-of-flight (TOF) lidar using a Geiger-mode avalanche photodiode (APD) by reducing range walk errors. Johnson et al. [22] derived the functional relationship that the received lidar echo signal waveform has with target orientation and other system parameters, studied the effects of mitigation strategies (such as increasing the transmitting beam size and reducing the pulse duration) on the range precision, and quantified the limitations of these methods. Zhou et al. [23] proposed the mean ranging accuracy and precision theory to objectively evaluate ranging performance. Luo et al. [24] proposed a cumulative pulse detection technique. Banakh [25] simplified the fine structure of a flow field by decomposing the flow field into a time-averaged flow field and a pulsating flow field and concluded that a shock wave will cause the beam to focus through numerical simulation. However, research based on numerical methods can only obtain the numerical solution under a single condition, which is not universal when evaluating the detection

accuracy of optical detection systems. Ref. [26] proposed a novel, to the best of our knowledge, compact, self-aligned focusing schlieren system that eliminates the need for a separate source grid and cutoff grid. Ref. [27] presents a type of single-shot ultrafast multiplexed coherent diffraction imaging technique to realize ultrafast phase imaging with both high spatial and temporal resolutions using a simple optical setup. These studies will better help the research and development of laser detection technology.

Most of the current research has focused on the influence of aero-optical effects on the optical passive imaging in the lateral optical windows of high-speed aircraft, and there is a lack of research on active laser detection in the forward optical windows. Therefore, based on the contributions of the above scholars, this paper further proposes the following research:

(1) To address the shortcomings of the conventional fixed-step ray tracing method, a new ray tracing method with adaptive step size adjustment is proposed.

(2) An aerodynamic environment simulation test of a missile-borne pulsed laser forward detection system at high flight speed is carried out. The aerodynamic environment of a munition flying at high speed is simulated by an intermittent transonic and supersonic wind tunnel to obtain the schlieren data of the flow field surrounding the missile-borne pulsed laser forward detection system at various Mach numbers. The schlieren data are compared with the simulation results for the flow field.

(3) The test equipment (such as the transonic wind tunnel) is used to simulate the flow field environment of the missile-borne pulsed laser forward detection system at a high flight speed. The aerodynamic characteristics of the model and the variation patterns of the pulsed laser echo waveform under different conditions are studied and compared with the simulation results of Reference [28]. The systematic and random errors of test data with Mach number show the same overall trends as those of the simulation data, which verifies the effectiveness of the semi-analytical method-based pulsed laser echo model.

## 2. Response Model

### 2.1. Equivalent Response Model of Missile-Fuze System

The propagation path of light in a medium mainly depends on the refractive index distribution of the medium. For a compressible gas medium, the factors affecting its refractive index include gas density, temperature, and composition. In general, the refractive index of the gas medium mainly depends on the density of the gas, and the Lorentz-Lorenz formula is typically used to describe the relationship between the refractive index and density of the flow field [1]:

$$\left(\frac{n^2-1}{n^2+2}\right) \cdot \frac{1}{\rho} = \frac{2}{3}K_{GD} \tag{1}$$

where $\rho$ is the flow field density, $n$ is the refractive index distribution of the flow field, and $K_{GD}$ is the Gladstone–Dale constant, which is generally considered to be related to the wavelength of light, as follows:

$$K_{GD} = 2.23 \times 10^{-4}\left(1 + \frac{7.52 \times 10^{-3}}{\lambda^2}\right) \tag{2}$$

In the formula, $\lambda$ is the wavelength of light, and its unit is μm; the unit for $K_{GD}$ is m$^3$/kg.

Although the refractive index of the atmosphere will change with air density, the overall refractive index of the atmosphere is still approximately 1 [29], so a simple approximation can be made to Formula (1): let $n^2 - 1 \approx 2(n-1)$, $n^2 + 2 \approx 3$; then, there is the following:

$$n = 1 + K_{GD} \cdot \rho \tag{3}$$

Based on Formula (3), the refractive index distribution can be obtained from the flow field density distribution.

From a mathematical point of view, the essence of the ray tracing problem in aero-optics is to solve the ray differential equation of the gas medium in the flowing state [25]:

$$\frac{d}{ds}\left(n(\bar{r})\frac{d\bar{r}}{ds}\right) = \nabla n(\bar{r}) \tag{4}$$

In the equation, $\vec{r} = x\vec{i} + y\vec{j}$ is the position vector of a point on the ray propagation path, $n(\vec{r})$ and $\nabla n(\vec{r})$ are the refractive index and the refractive index gradient at the point, respectively, and $s$ is the ray propagation path. This equation basically has no analytical solution for the light propagation path in any nonuniform refractive index field. The equation has analytical solutions only in some special cases, such as a spherical, cylindrical, or planar iso-refractive index surface. In these cases, the equation is generally solved by numerical methods.

### 2.2. Improved Ray Tracing Method Based on Adaptive Step Size Adjustment

The accuracy of ray tracing based on the Runge-Kutta method is affected by the tracing step size. The conventional fixed step size method uses a fixed tracing step size $\Delta s$. The smaller the value of $\Delta s$ is, the higher the tracing precision, but the computational load will increase at the same time. For a complex high-speed nonuniform flow field, a constant step size cannot meet the tracing precision requirements in areas with gentle or severe refractive index variations and will result in a waste of computing resources. Therefore, the tracing step size is adaptively adjusted so that the step size is larger in places where the refractive index changes gently and smaller in places where the refractive index changes sharply. This method not only can ensure accuracy but also can improve the computational efficiency.

The refractive index gradient essentially reflects the magnitude of the refractive index variation, so light propagation is most sensitive to the refractive index gradient. In this section, a step size adjustment method based on the refractive index gradient is proposed. The method essentially constructs a step size adjustment function with the maximum refractive index gradient of the grid node as a variable. The following piecewise function is given through the previous pilot calculation:

$$f(\Delta n) = \begin{cases} l_i, & \Delta n < 10^{-5} \\ 0.5l_i, & 10^{-5} < \Delta n \le 10^{-4} \\ 0.2l_i, & 10^{-4} < \Delta n \le 10^{-3} \\ 0.1l_i, & 10^{-3} < \Delta n \le 10^{-2} \\ 0.05l_i, & \Delta n > 10^{-2} \end{cases} \tag{5}$$

In the formula, the tracing step length $\Delta s = f(\Delta n)$, $\Delta n = n_{\text{gmax}} - n_{\text{gmin}}$ is the difference between the maximum and minimum refractive index values of the neighboring grid nodes of the calculating position, $l_i = \sqrt{a_i \times b_i}$ is the average geometric size of grid cells in the calculating position $i$, and $a_i$ and $b_i$ are the grid cell side lengths.

There are two explanations for Formula (5):

(1)  $l_i$ is set to not exceed the average geometric size of the local grid cells. The reason is that considering the turbulent phenomenon of sharp changes in the density of shock waves in the shock-wave flow field, the accuracy of computational fluid dynamics (CFD) in capturing shock waves is generally two to three grid cells. Although the refractive index changes greatly in these grids, the change in the refractive index in the local area before and after the shock wave may be very gentle; if $l_i$ is set too large, it may cross the shock wave structure and cause a large error in the result.

(2)  The minimum tracing step size is 0.05 $l_i$, which is the optimal value determined by multiple numerical tests.

For the high-speed nonuniform flow field studied in this paper, since an accurate refractive index distribution function cannot be obtained, the exact solution of the light

propagation path cannot be obtained. To verify the correctness and effectiveness of the ray tracing method proposed in this paper, ray tracing in a medium with a radial refractive index gradient distribution is performed by the conventional fixed-step ray tracing method and the adaptive step size adjustment method proposed in this paper. The tracing results are compared with the analytical solutions to analyze the advantages and disadvantages of the two methods.

Since the calculation results of CFD software are based on the discrete data of irregular quadrilateral grid nodes, the refractive index and refractive index gradient of any point cannot be directly obtained. Therefore, it is necessary to interpolate the number of any point in the grid. Considering the interpolation accuracy and time complexity, this paper adopts the inverse distance weighted average interpolation method based on the quadrilateral grid with good stability and accuracy.

The position of the first step in the ray tracing process is known as $P = (x_i, y_i)$, and the nearest four mesh vertices are found and numbered 1, 2, 3, 4. Then, based on the refractive index of the four vertices $n_{1,2,...4}$, the refractive index and refractive index gradient are interpolated at $P$ as follows:

$$n_P = \frac{\sum\limits_{i=1}^{4} \left( n_i \prod\limits_{\substack{j=1 \\ j \neq i}}^{4} d_j \right)}{\sum\limits_{i=1}^{4} \prod\limits_{\substack{j=1 \\ j \neq i}}^{4} d_j} \tag{6}$$

where $d_i = \sqrt{(x - x_i)^2 + (y - y_i)^2}, i = 1, 2, \ldots 4$ is the distance from the point $P$ to the four vertices of the grid.

Given the refractive index of space grid nodes, the refractive index gradient at the grid nodes should be solved first, and then the gradient value of any point in the grid can be obtained by the interpolation method.

For a medium with a known refractive index distribution of the flow field, the refractive index gradient can be obtained by taking the derivative of the refractive index distribution function. In this paper, the gradient operator is used to solve the grid point refractive index gradient, and the Barron gradient operator is selected to solve the refractive index gradient.

The Barron gradient operator is the result of the cubic spline difference of discrete refractive index values at five points in the neighborhood of the desired node, and the refractive index gradient at the node can be obtained as follows:

$$\begin{cases} \frac{\partial n(i,j)}{\partial x} = \frac{1}{12}n(i-2,j) - \frac{8}{12}n(i-1,j) + \frac{8}{12}n(i+1,j) - \frac{1}{12}n(i+2,j) \\ \frac{\partial n(i,j)}{\partial y} = \frac{1}{12}n(i,j-2) - \frac{8}{12}n(i,j-1) + \frac{8}{12}n(i,j+1) - \frac{1}{12}n(i,j+2) \end{cases} \tag{7}$$

Figure 1 shows the tracing errors at different tracing step lengths. It can be seen in the figure that the tracing precision is not much different among the cases where the step lengths are 0.5 mm and 0.2 mm and adaptive. The tracing error at the adaptive step length is slightly smaller than those at the other two step sizes, which are all on the order of $10^{-4}$. The reason for this phenomenon is that the refractive index is spatially discretely distributed, and the refractive index and refractive index gradient during the tracing process are obtained by interpolation, which inevitably introduces errors. Notably, the refractive index gradient needs to be interpolated twice, which eventually leads to a larger computation error. Comparing the curves at the three step length settings, the tracing error increases with increasing z-axis distance, which is accompanied by oscillation, and

the maximum error at the adaptive step size is $3.09 \times 10^{-4}$. In addition, the number of steps required for tracing decreases with increasing tracing step size, and the adaptive step size adjustment method has the fewest tracing steps, so its computational efficiency is high. The results show that for the same grid cell, the ray tracing error is not sensitive to the tracing step size, but the adaptive step size adjustment ray tracing method can significantly improve the tracing efficiency.

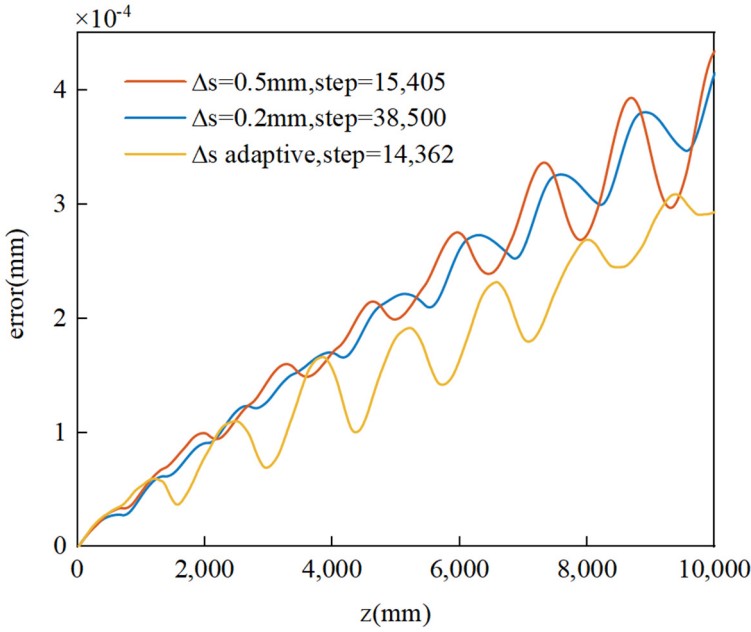

**Figure 1.** Relative error of ray tracing along the z-axis.

In summary, the adaptive step size adjustment ray tracing method proposed in this paper has high computational accuracy and can effectively reduce the number of tracing steps, thereby reducing the computational load. The proposed method is suitable for ray tracing of any medium with a discrete spatial distribution of the refractive index.

## 3. Test Model and Test Method

In this paper, the test equipment (such as the transonic wind tunnel) is used to simulate the flow field environment of a missile-borne pulsed laser forward detection system at high flight speed and studies the aerodynamic characteristics of the model and the variation patterns of the pulsed laser echo waveform under different conditions. The model of the missile-borne pulsed laser forward detection system studied in this paper is shown in Figure 2. Due to the straight ballistic trajectory of the high-explosive anti-tank weapon, the angle of attack can be considered zero.

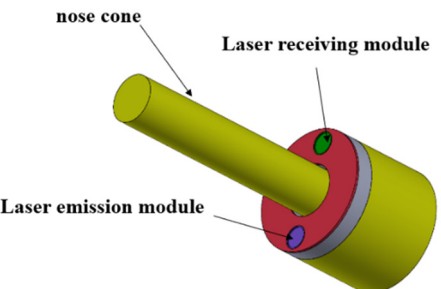

**Figure 2.** Geometric model of the missile body.

The typical working scene of the missile-borne pulsed laser forward detection system is the complex battlefield environment, which puts forward high requirements on the laser, which needs to meet the requirements of small volume, high repetition frequency, high output power, and working stability. After comprehensive consideration of device volume and performance, we chose an SPLPL90-3 semiconductor laser produced by OSRAM company, which has the advantages of having a small size, low power consumption, and long service life. The laser adopts a series laminated welding assembly mode, plastic packaging, and a small emission angle (<25°), and the output laser wavelength is 905 nm. The spectral range is 40 nm, and the output peak power is 75 W.

### 3.1. Test Equipment

The tests in this paper were all carried out in a $0.6 \times 0.6$ m intermittent transonic and supersonic wind tunnel at the China Aerodynamics Research and Development Center. The wind tunnel (shown in Figure 3) is a DC intermittent transonic and supersonic wind tunnel. The cross-sectional dimensions of the wind tunnel test section are $0.6 \times 0.6$ m, and the test section is 2.5 m long. At a transonic speed, the top and bottom walls are 60° effusion walls, the porosity of the model area is 4.3%, the porosity of the acceleration area is 4.4%, and the left and right walls are solid walls; at supersonic speed, the four walls are all solid walls.

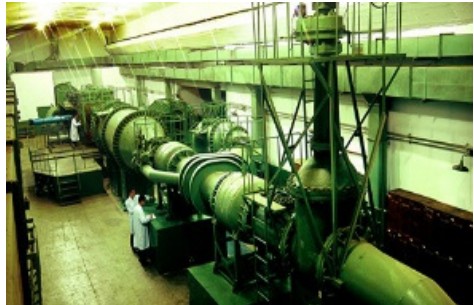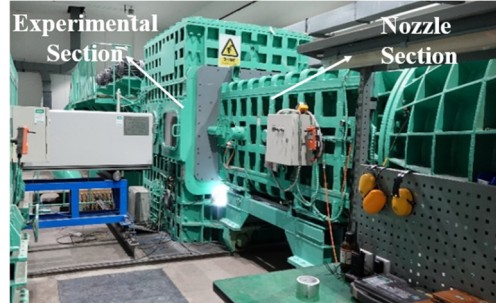

**Figure 3.** Wind tunnel performance indexes.

The test model can be placed inside the wind tunnel. The wind tunnel has a wide observation range for the spatial flow field and a large range of test angles of attack. The Mach number is adjusted by replacing the binary spray nozzle. The height of the supersonic diffuser can be adjusted. The wind tunnel also contains an adaptive test section. The main technical performance indexes of the $0.6 \times 0.6$ m intermittent transonic and supersonic wind tunnel are as shown Table 1.

**Table 1.** Wind tunnel performance indexes.

| Parameter | Numerical Value |
| --- | --- |
| Mach number | 0.4~4.5 |
| Attack angle | $-1° \sim 49°$ |
| Angle of sideslip | $-7°, -5°, -3°, -1°, 0°, 1°, 2°, 3°, 5°$ |
| Stagnation pressure | $(0.95 \sim 7.2) \times 10^5$ Pa |
| Dynamic pressure | $(0.056 \sim 0.69) \times 10^5$ Pa |
| Reynolds number (1/m) | $(9.87 \sim 28.7) \times 10^6$ |
| Total temperature | 280~290 K |

### 3.2. Measurement of Aerodynamic Characteristics of the Supersonic Flow Field

In the aerodynamic flow field measurement test, the test model is the equivalent model of the missile-borne pulsed laser forward detection system, and the blockage ratio of the wind tunnel at a 0° angle of attack is approximately 2%, which meets the requirements of the aerodynamic test.

The model is supported on the bottom surface of the wind tunnel test section by a special bracket and is adjusted along the z direction so that the model is exactly at the center of the observation window. To minimize the influence on the flow field, the windward side of the bracket is designed in the shape of a knife edge, and the distance between the model axis and the bottom surface is 330 mm.

This test aims to obtain the internal structure of the fluid near the model head in the supersonic flow field. A 250 mm hybrid schlieren system is used to visualize the stable flow field. The test conditions are set as shown in Table 2.

**Table 2.** Test conditions.

| Parameter | Numerical Value |
|---|---|
| Mach number | 2.0, 3.0, 4.0 |
| Attack angle (°) | 0 |
| Total temperature $T_0$ (K) | 288 |
| Total pressure $P_0$ (MPa) | 0.19, 0.36, 0.63 |
| Density $\rho$ (kg/m$^3$) | 0.51, 0.33, 0.20 |
| Time (s) | >10 s |

The test steps are as follows:

(1)  Fix the test model on the bracket, adjust the position of the bracket so that the model is in the center of the schlieren observation window, and then fix the model.
(2)  Configure a 250 mm hybrid schlieren system to observe whether the display effect is normal under static conditions.
(3)  Turn on the air inlet switch of the wind tunnel, release the high-pressure gas, and record the data at the corresponding Mach number after the flow field is stable.
(4)  Turn off the air inlet switch, and check whether the test model is in good condition after the airflow disappears and whether the optical lens is loose, falling off, or deformed.
(5)  Replace the nozzle with a nozzle with a different Mach number, repeat the above steps, and finally obtain the schlieren data of the flow field under 2–4 Ma.
(6)  After the test, turn off the electrical equipment and record and process all test data.

*3.3. Echo and Ranging Data Collection by the Missile-Borne Pulsed Laser Forward Detection System*

The laser pulses emitted by the missile-borne pulse laser forward detection system are scattered by objects with a certain reflectivity before the laser receiving module can receive the echo signal. However, the wind tunnel used in the test does not have qualified physical objects in the test section or the nozzle section and thus cannot meet the test requirements for receiving echo signals. To solve this problem, this paper proposes a test plan for installing a reflector in an air duct.

The reflector is installed on the upper part of the structure; its size is 120 mm × 90 mm according to the emitted laser spot, and its reflectivity is 0.3. The reflector is connected to the wind tunnel strut through a special bracket. To reduce the wind resistance and minimize the impact on the flow field, the windward surface of the bracket is designed in the shape of a knife edge.

The test model is the principal prototype of the missile-borne pulsed laser forward detection system, and its structural dimensions are basically the same as the equivalent model in the flow field measurement test. The reflector is connected to the wind tunnel strut through a bracket. The laser pulse emitted by the missile-borne pulse laser forward detection system irradiates the reflector, and the scattered echo signal is captured by the receiving module. The plane on which the laser emitting and receiving module of the model is located is 1.5 m apart from the reflector.

For an extended plane target with an inclination angle, the target plane is at an angle to the *xoy* plane, and then $\varphi = \beta$. The acquirability is as follows:

$$P(t, R) = \frac{P_0 D^2}{2\omega^2 R^2} \eta_{atm}^2 \eta_{sys} \iint_A e^{-2\frac{(x^2+y^2)}{\omega^2}} e^{\frac{-(t-\frac{2R+2y\tan\beta}{c})^2}{\tau^2}} f_r(\beta) \cos\beta \, dxdy \tag{8}$$

The laser beam is deflected when passing through the high-speed non-uniform flow field, and then the actual propagation path of the first ray of the laser transmission process can be determined according to the following formula:

$$R_{Li} \approx \frac{(R_r - l)}{\cos\psi_i} + OPL_i \tag{9}$$

where $l$ is the length of the nose cone of the head of the projectile, $\psi_i$ is the deflection angle between the light and the axis $z$ caused by the aerodynamic optical effect, which can be obtained by the ray tracing method, and $OPL_i$ is the actual path of the light through the non-uniform flow field

After the light is deflected in the high-speed flow field, the incidence angle when it reaches the target plane is as follows: $\varphi_i' = \beta + \psi_i$.

For extended targets, because the size of the target is much larger than the radius of the laser beam, and the deflection of the beam caused by the shock wave is small during short-range detection, it can be considered that all the transmitted beams reach the target plane. In the receiving process, wavefront distortion occurs when the beam passes through the non-uniform flow field, resulting in attenuation of the intensity of the beam in the pupil plane. The Strehl ratio can be used to characterize the ratio of light intensity in the spot on the target surface with or without aberration, that is, the ratio of spot energy with or without the interference of aero-optics. The pulsed laser echo equation affected by aero-optics can be expressed as follows:

$$P_r'(t) = \frac{P_0 D^2 \eta_{atm}^2 \eta_{sys}}{2\omega^2} \iint_\Sigma e^{-2(\frac{x^2+y^2}{\omega^2})} e^{-\frac{(t-\frac{2R_L}{c}-\frac{2y\tan\beta}{c})^2}{\tau^2}} \cdot \frac{f_r(\varphi')\cos\varphi'}{R_L^2} \cdot SR \, dxdy \tag{10}$$

where this SR denotes the Strehl ratio at which the echo beam reaches the pupil of the receiving system, which can be obtained by the ray tracing method:

$$P_r'(t) == \frac{\sqrt{\pi} P_0 D^2 \eta_{atm}^2 \eta_{sys}}{2\sqrt{2}\omega} \int_{-\omega}^{\omega} e^{\frac{-2y^2}{\omega^2}} e^{-\frac{(t-\frac{2R_L}{c}-\frac{2y\tan\beta}{c})^2}{\tau^2}} \cdot \frac{f_r(\varphi')\cos\varphi'}{R_L^2} \cdot SR \, dy \tag{11}$$

Since the SR $R$ and $\varphi'$ sum of each ray in the emitted beam are different, the above equation cannot be integrated directly. Based on the principle of geometric optics, the beam is regarded as multiple thin beams that propagate independently in the flow field, and the echo power can be obtained by superimposing the power of each ray on the entry pupil plane:

$$P_r'(t) = \frac{\sqrt{\pi} P_0 D^2 \eta_{atm}^2 \eta_{sys}}{2\sqrt{2}\omega} \sum_{i=1}^{M} e^{\frac{-2y_i^2}{\omega^2}} e^{-\frac{(t-\frac{2R_{Li}}{c}-\frac{2y_i\tan\beta}{c})^2}{\tau^2}} \cdot \frac{f_r(\varphi_i')\cos\varphi_i'}{R_{Li}^2} \cdot SR(i) \cdot \Delta y_i \tag{12}$$

where $M$ is the number of tracing rays. The above equation is the power equation of pulsed laser echo affected by the aero-optic effect.

To obtain the real-time pulsed laser echo signal when the wind tunnel blows air, the received echo signal is transmitted to the host computer through the wire, and the corresponding waveform data are recorded. The test installation diagram is shown in Figure 4.

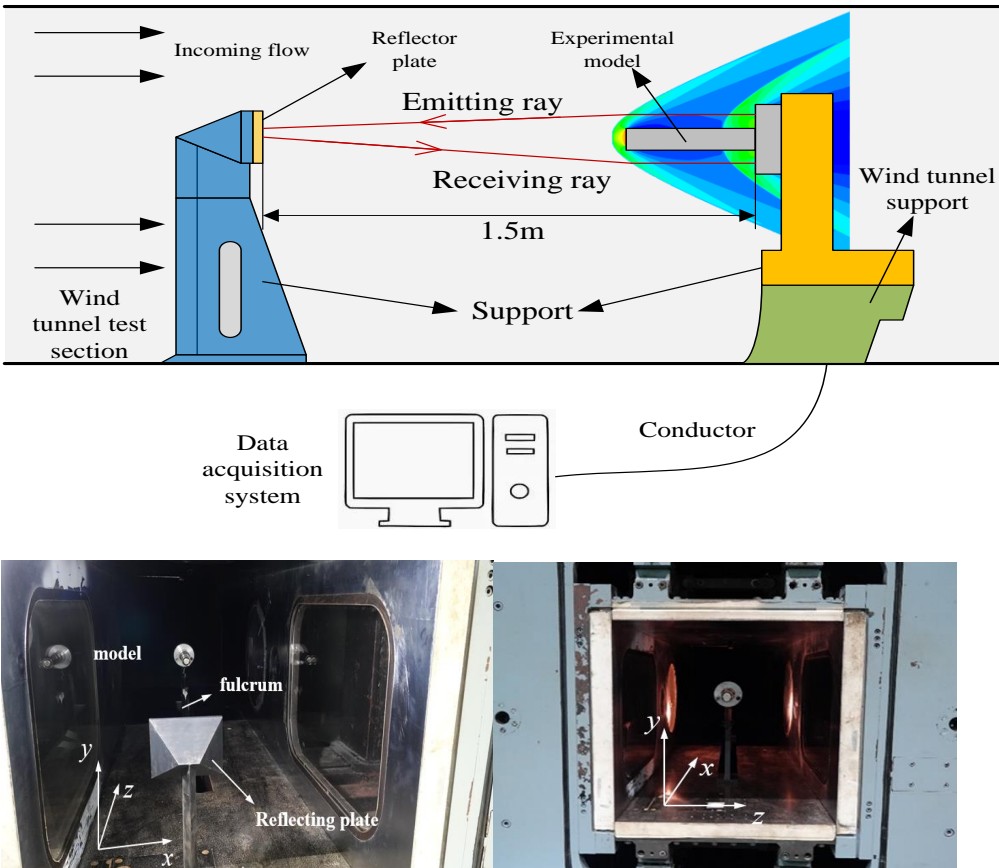

**Figure 4.** Schematic diagram of the test layout.

The working conditions of the test are the same as those shown in Table 2. The corresponding test steps are as follows:

(1) Fix the test model and the reflector on the brackets, install the two brackets on the wall surface and the strut of the wind tunnel test section, respectively, and adjust the distance between the laser emitting module and the reflector to 1.5 m.

(2) Connect the test model to the external power supply and the data acquisition system through wires, turn on the power to observe whether the emitted laser spot is located in the center of the reflector, and record the laser echo signal and ranging data in static conditions.

(3) Turn on the air inlet switch of the wind tunnel, release the high-pressure gas, and record the echo signal and ranging data at the corresponding Mach number after the flow field is stable.

(4) Turn off the air inlet switch and check whether the test model is in good condition after the airflow disappears and whether the wires are broken or deformed.

(5) Replace the nozzles with different Mach numbers, repeat the above steps, and finally obtain the laser echo signals and ranging data at Mach 2, 3, and 4.

(6) After the test, turn off the electrical equipment and record and process all test data.

## 4. Analysis of Test Results

### 4.1. Visualized Schlieren Analysis of the Supersonic Flow Field

The schlieren results of the external flow field of the test model at various Mach numbers are shown in Figure 5. Due to the limitation of the field of view of the schlieren system, it is impossible to obtain the complete data of the external flow field for the model at one time, so imaging is performed twice. The left and right figures correspond to the schlieren images of the front and rear of the test model, respectively.

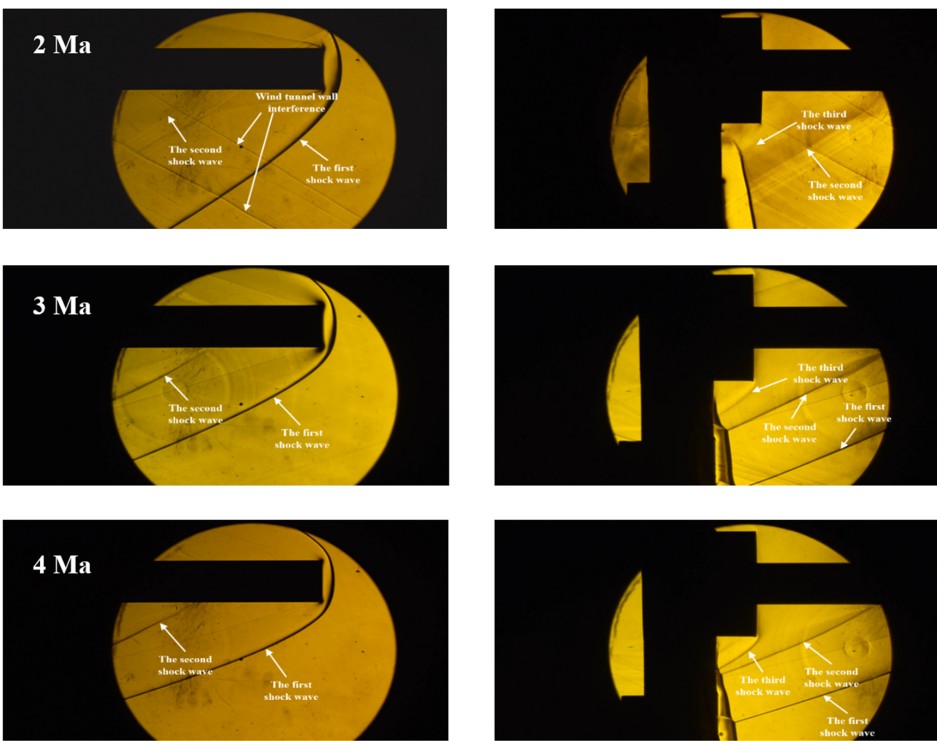

**Figure 5.** Schlieren diagrams of the flow field at the model front and rear.

As seen in Figure 5, when the flow field is stable, three shock waves will be generated around the model. The first shock wave is located directly in front of the model head, which is a detached shock wave; the second shock wave starts to extend from the middle part of the nose cone, which is caused by the blockage of the airflow by the step structure in the model rear; the third shock wave is located at the corner of the step surface, and the airflow blocked by the step surface is accelerated to supersonic speed here, which is the reason for the generation of the third shock wave.

Overall, with increasing Mach numbers, the intensity of the shock wave increases, that is, the density difference before and after the shock wave also increases. The phenomenon is most obvious from 2 Ma to 3 Ma. The second and third shock waves are weaker at 2 Ma, and no clear structure can be observed at this time, but when the Mach number increases to 3, the structures of the two shock waves become very clear. In addition, the shock angle decreases with increasing Mach numbers.

Figure 6 compares the test and simulated flow field structures. The upper half is the simulated flow field, and the lower half is the test flow field. Comparing the flow field structures at different Mach numbers, the simulation results of [30] match the test results well, and the test and simulated shock wave structures are relatively consistent. There is a certain error between the test and simulated shock angles due to the difference in gas density used in the simulation and tests. The high-speed airflow in the wind tunnel is generated by compressing the gas and then expanding it. The gas density in the test section decreases rapidly with increasing Mach numbers. The gas density at 4 Ma is only 0.2, while the density of ideal gas is used in simulation, so the difference between the two is large, which results in a certain deviation in the results. In general, the above analysis fully verifies the correctness and effectiveness of the flow field simulation model, and the relevant simulation results can be used for an alternative analysis of the wind tunnel test.

### 4.2. Analysis of the Detected Echo

The test echo waveforms at different Mach numbers are shown in Figure 7. The echo peak value gradually decreases with increasing Mach numbers. As shown by the echo peak voltages corresponding to the different Mach numbers in Table 3, the echo peak voltage

drops by 0.281 from 0 to 4 Ma. According to the reference analysis [1], the precision of pulsed laser ranging is significantly related to the rising edge of the laser echo waveform. To clearly show the rising edge of the echo waveform at each Mach number, the data in the black box are enlarged, as shown in the inset. The rising edge of the echo shows a trend of a continuous right shift with increasing Mach numbers, which will affect the ranging accuracy.

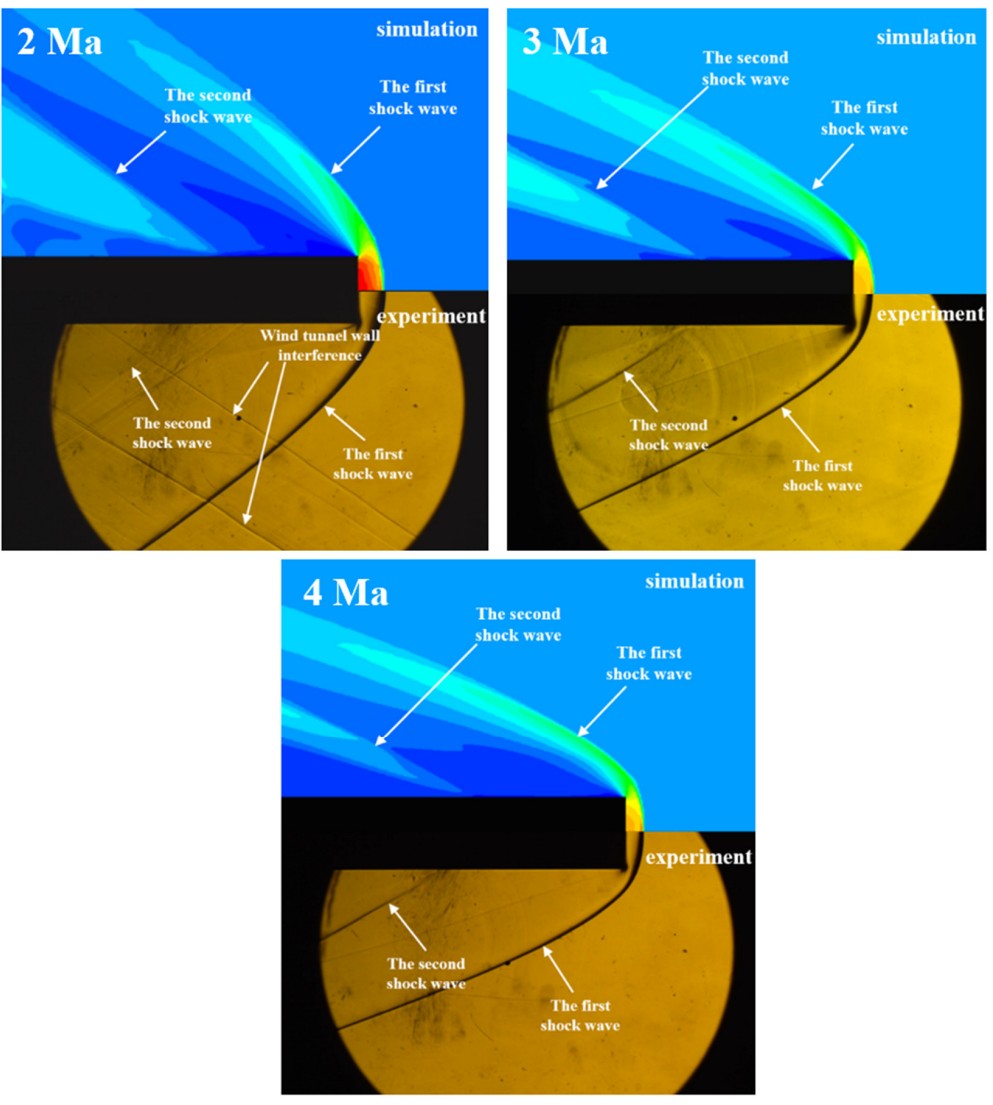

**Figure 6.** Comparison of schlieren diagrams of the flow field.

**Table 3.** Test echo peak voltages at different Mach numbers.

| Mach Number | 0 Ma | 2 Ma | 3 Ma | 4 Ma |
|---|---|---|---|---|
| Test echo peak voltage (V) | 2.019 | 1.878 | 1.809 | 1.738 |

The test echoes are compared with the theoretical echo waveforms at different Mach numbers, as shown in Figure 8. The test and theoretical echo waveforms are in good agreement under static conditions. With increasing Mach numbers, the difference between the test and simulated echo waveforms gradually increases, which is manifested as the peak value of the test echo being larger than that of the theoretical echo. At 4 Ma, the difference between the test and theoretical peak values reaches 0.43 V, and the pulse width of the test echo also expands. Table 4 lists the errors of the test echo peak values relative

to the simulated echo peak values at different Mach numbers. The relative error of the two peak values is the smallest under static conditions, at only 1.28%. With increasing Mach numbers, the relative error of the two peak values increases significantly and reaches 18.65% at 4 Ma. The reason is that the wind tunnel cannot completely simulate the real ballistic flight environment. In a confined space with a limited air duct, the gas density and temperature are obviously lower than those in a real ballistic flight environment. Due to the large difference from the actual free space flight environment of the munition, there is a certain difference between the test and simulated results.

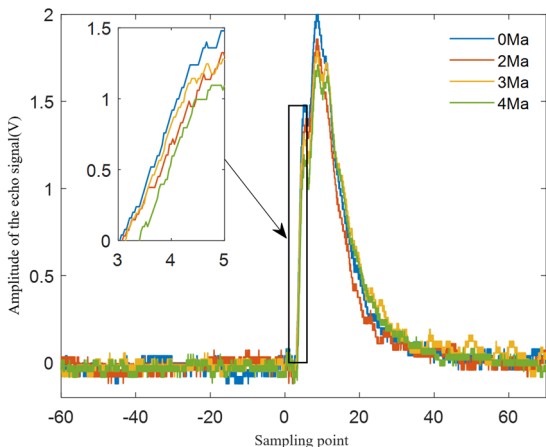

**Figure 7.** Comparison of test echo waveforms at different Mach numbers.

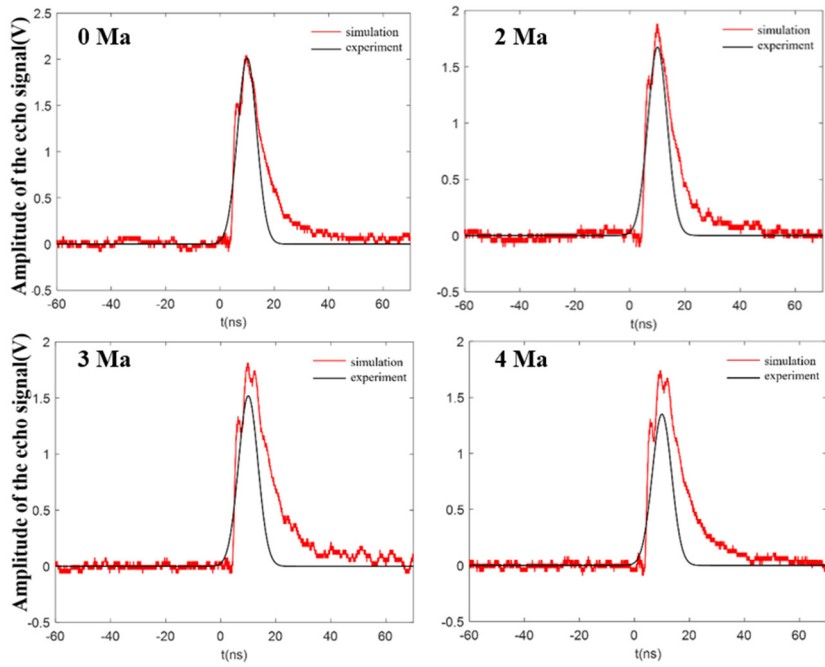

**Figure 8.** Test and simulated echo waveforms at different Mach numbers.

**Table 4.** Relative errors of the test and simulated echo peak values at different Mach numbers.

| Mach Number | 0 Ma | 2 Ma | 3 Ma | 4 Ma |
|---|---|---|---|---|
| Peak relative error | 1.28% | 8.51% | 13.87% | 18.65% |

To further study the correlation between the test and theoretical echo signals, the variable elements of the two are analyzed by the correlation principle to measure the

closeness of the correlation between the two variables. The correlation coefficient is a statistical indicator of the degree of linear correlation between variables, usually denoted as $r$. If there are two sets of discrete variables $X$ and $Y$, whose data length is $n$, then there is a correlation coefficient matrix:

$$R_{xy} = \frac{Cov(X,Y)}{\sqrt{Cov(X,X)Cov(Y,Y)}} \tag{13}$$

where $Cov(X,Y)$ is the covariance of $X$ and $Y$, which satisfies the following:

$$Cov(X,Y) = E\left[\left(X - \frac{1}{n}\sum_{i=1}^{n} X_i\right)\left(Y - \frac{1}{n}\sum_{i=1}^{n} Y_i\right)\right] \tag{14}$$

With only two column vectors, $R_{xy}$ is a $2 \times 2$ symmetric matrix, its main diagonal elements are 1, and the sub-diagonal elements are the correlation coefficients of the two column vectors ($-1 < r < 1$).

Although the statistical community has not developed a unified control standard for the significance of the data represented by the correlation coefficient, the correspondence between the correlation coefficient and the degree of correlation can be defined by referring to Table 5.

**Table 5.** Correspondence between the correlation coefficient and the degree of correlation.

| Correlation Coefficient | Degree of Correlation |
|:---:|:---:|
| 0.00~±0.30 | Micro correlation |
| ±0.30~±0.50 | Real correlation |
| ±0.50~±0.80 | Significant correlation |
| ±0.80~±1.00 | Highly correlated |

According to the correlation principle, the test and simulated data are substituted into Formula (7) for calculation. The correlation coefficients at different Mach numbers are shown in Table 6.

**Table 6.** Correlation coefficients of test and simulated echo data at different Mach numbers.

| Mach Number | 0 Ma | 2 Ma | 3 Ma | 4 Ma |
|:---:|:---:|:---:|:---:|:---:|
| Correlation coefficient | 0.7981 | 0.7745 | 0.7561 | 0.7432 |

The results show that the correlation coefficients of the test and simulated echo signals under the four working conditions all exceed 0.7, indicating a significant correlation. Moreover, the correlation coefficient between the two signals is the largest under static conditions, which indicates that the two signals have the highest degree of correlation at this time, which is close to a high correlation.

In general, the simulated echo ranging errors at different Mach numbers are roughly consistent with the changing trend of the test data. This is due to the limitation of the test conditions. These are, for example, the support interference, which means the support will cause interference to the simulated flow field, and the boundary interference, which means the wind tunnel flow field has a boundary but the real atmosphere has no boundary. These disturbances are the reasons for the limitation. The environment inside the wind tunnel is notably different from the free space environment of the actual outer ballistic trajectory. The gas temperature and density in the simulation are significantly lower than those in the actual environment, which results in a certain difference between the peak values and shapes of the simulated and test echo waveforms, which is ultimately reflected in the partial deviation of the ranging error between the simulation and the test.

## 5. Conclusions

In this paper, a simulation test of the aerodynamic environment of the missile-borne pulsed laser forward detection system at high flight speed was conducted. By designing a new adaptive step size adjustment method, the calculation efficiency was greatly improved. At the same time, the high-speed wind tunnel was used to simulate the high-speed flight state, and the real schlieren data of the flow field were obtained.

The conclusions are as follows:

(1) A new ray tracing method with adaptive step size adjustment is proposed. By setting a step size adjustment piecewise function, the gradient change of the refractive index inside the grid cells can be captured, and the step size in the ray tracing process can be adaptively controlled. The results show that although the error of ray tracing is not sensitive to the tracing step size in the same grid cell, the adaptive step adjustment ray tracing method can significantly improve the tracing efficiency.

(2) In a simulation test of the aerodynamic environment of the missile-borne pulsed laser forward detection system at a high flight speed, the aerodynamic environment of the munition at high flight speed is simulated by the intermittent transonic and supersonic wind tunnel to obtain the schlieren data of the flow field surrounding the missile-borne pulsed laser forward detection system at various Mach numbers. The schlieren data are compared with the simulation results of the flow field. The results show that the two match well and present similar shock wave structures, which verifies the correctness and effectiveness of the flow field simulation model.

(3) The echo waveform measurement of the missile-borne pulsed laser forward detection system is completed. The laser echo signals at different Mach numbers are sampled, and the test echo data are compared with the simulated echo data. The results show that the peak value of the test echo decreases gradually with increasing Mach numbers and drops by 0.281 V from 0 to 4 Ma. The test echo waveforms are in good agreement with the simulated echo waveforms, and the relative error between the peak values of the test and simulated echo waveforms at each Mach number does not exceed 20%. In addition, the correlation coefficients of the test and simulated echo waveforms all exceed 0.7, indicating high correlations between the two.

**Author Contributions:** Conceptualization, P.L.and J.L.; methodology, T.H.; software, P.L.; validation, P.L., J.L. and T.H.; formal analysis, H.Z.; investigation, P.L.; resources, H.Z.; data curation, J.L.; writing—original draft preparation, P.L.; writing—review and editing, J.L.; visualization, H.Z.; supervision, H.Z.; project administration, T.H.; funding acquisition, H.Z. All authors have read and agreed to the published version of the manuscript.

**Funding:** This research received no external funding.

**Institutional Review Board Statement:** Not applicable.

**Informed Consent Statement:** Not applicable.

**Data Availability Statement:** Data is unavailable due to privacy.

**Conflicts of Interest:** The authors declare no conflict of interest.

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
