# Peer review of "Simulation Test of The Aerodynamic Environment of A Missile-Borne Pulsed Laser Forward Detection System at High Flight Speed"

_photonics, doi:10.3390/photonics10121363_

Round 1

Reviewer 1 Report

Comments and Suggestions for Authors

The overall structure of the article is complete, with clear descriptions of key parts. However, the level of English writing can be further improved, and some details need to be improved. The details are as follows:

1. Further improve the reference literature, pay attention to the uniformity of reference format, and supplement the latest research in relevant fields;

2.Figure 3 shows the overall experimental equipment, but the labeling of the experimental equipment is not clear;

3.Figure 4 provides an explanation of the on-site arrangement of the experiment, which is very good and concise. Regarding the experimental section, it is recommended to add some pictures of the actual interior of the wind tunnel;

4.The conclusion that the refractive index of air is close to 1 at different densities, as stated in lines 136 and 137 of the article, needs to be supplemented with evidence or references.

5.In lines 184 to 191 of the article, please describe the specific form of the radial refractive index gradient distribution in the simulation conditions. Additional experimental details are required.

6.In line 352, I can’t find the specific context or content surrounding the phrase "According to the above analysis", Could you please provide more information or clarify the context in which this phrase appears?

7.Lines 366-367, the data in the table does not correspond to the image. The peak discrepancy in Figure 8 at 4 Mach speed did not reach 18%, and the peak discrepancy in the 2 Mach figure is significantly larger than at 4 Mach. It is necessary to check whether there are errors in the chart data.

8.The conclusion section is divided into three points to express the overall contribution of the article, but lacks a comprehensive conclusion. It is recommended to further refine the three sub items to highlight the overall contribution of the article.

Comments on the Quality of English Language

There is still room for further improvement in the author's English writing proficiency

Author Response

Dear reviewer:

1-8 suggestions have been made in the comments of the article,See the attachment for the revised article.

Reviewer 2 Report

Comments and Suggestions for Authors

The submitted work studies modelling of the aero-dynamic environment of a missile-borne pulsed laser forward detection system at high flight speed. These are my comments thereupon.

1. Although the word ‘laser’ is used in the title, the Authors do not provide a description of its radiation parameters. It is not clear, for which laser modelling was performed. This laser’s output power, beam parameters (diameter, divergence), pulse properties (duration, shape, repetition rate) should be accounted for in modelling. If the Authors refer to a specific laser, its model number should be given, as well.

2. The Conclusion summarises the modelling results at different various Mach numbers. It is equally (or, maybe, even more) interesting, however, how the results depend upon the radiation parameters (power and wavelength of the radiation, &c). A corresponding analysis needs to be added.

Once the offered comments are addressed in a revised version of this manusctipt, it may be published in Photonics.

Author Response

Dear reviewer:

1-2 suggestions have been made in the comments of the article,See the attachment for the revised article.

Reviewer 3 Report

Comments and Suggestions for Authors

The present manuscript analyzes the difficulties related to laser detection in a high-velocity setting. It puts forward a novel approach utilizing a ray tracing technique to address these challenges. The manuscript further conducts simulations to replicate the aerodynamic conditions and subsequently verifies the accuracy of the simulation outcomes through experimental testing. The results demonstrate a significant degree of correspondence between the simulated and experimental data, thereby establishing a strong correlation. If all of these problems (mentioned in the attached file) can be effectively addressed, this research has the potential to be published in MDPI Photonics:

Author Response

Dear reviewer:

1-11 suggestions have been made in the comments of the article,The overall format of the article has been further improved.See the attachment for the revised article.

Round 2

Reviewer 2 Report

Comments and Suggestions for Authors

In response to my observations, important information was added to the manuscript that made it more interesting and comprehensible. My comments have been fully addressed by the Authors in the revised manuscript, which may be now published.

Reviewer 3 Report

Comments and Suggestions for Authors

I thank the authors for their replies, and the replies satisfy me. I would like to recommend the publication of the current manuscript in the journal.